# Identification of an optimal exogenous gene insertion site (P–M) and establishment of a reverse genetics system for aMPV/A

Yu Guo[1,2,3,4*], Ting Xue[1,2], Bingqing Lu[1,2], Jiqing Zheng[1], Yirong Wang[1], Xueqiao Xie[1], Huiting Wang[1], Furui Cao[1], Yijun Zhang[1], Jianhua Wang[1*]

1 The College of Biomedical Medicine, Suzhou Chien-Shiung Institute of Technology, Suzhou, China, 2 Suzhou Chien-Shiung Biopharmaceutical Industry Think Tank, Suzhou, China, 3 Institute of Animal Husbandry and Veterinary Medicine, Beijing Academy of Agriculture and Forestry Sciences, Beijing, China, 4 The College of Veterinary Medicine, Hebei Agricultural University, Baoding, China

* 9207@csit.edu.cn (JW), 0514@csit.edu.cn (YG)

## Abstract

Avian metapneumovirus subgroup A (aMPV/A) causes widespread infections in commercial poultry worldwide. The development of highly efficient viral vector vaccines is constrained by multiple factors, among which the lack of a mature reverse genetics system and the uncertainty regarding the optimal insertion site for foreign genes represent one of the core technical bottlenecks. Here, we constructed an aMPV/A reverse genetics system based on the pBR322 vector and inserted EGFP into seven of its intergenic regions to generate recombinant viruses. Viral replication, exogenous gene expression and genetic stability were systematically assessed via multiple assays. The results demonstrated that the expression level at the P-M locus after EGFP insertion was approximately 1.2-fold higher than that of other recombinant viruses. In addition, there was no significant difference in viral titer, and good genetic stability was observed over 20 generations. This study establishes a stable and efficient reverse genetics system for aMPV/A, which provides a critical technical platform for aMPV/A molecular mechanism research and novel recombinant vector vaccine development, with important practical value for global aMPV prevention and control.

## 1. Introduction

Avian metapneumovirus (aMPV) poses a significant threat to the global poultry industry, causing substantial economic losses [1]. aMPV belongs to the family Pneumoviridae and genus Metapneumovirus, and is a single-stranded negative-sense RNA virus [2]. Its genome harbors eight genes encoding nine proteins, namely nucleoprotein (N), phosphoprotein (P), matrix protein (M), fusion protein (F), M2 protein (including M2-1 and M2-2 isoforms), small hydrophobic protein (SH), attachment protein (G), and large polymerase protein (L) [3]. According to genomic sequence and antigenic variations, aMPV is currently classified into four major subtypes (A, B, C, and D),

**Data availability statement:** All relevant data are within the manuscript and its Supporting Information files.

**Funding:** This work was financially supported by Start-up fund for New Ph.D. Researchers of Suzhou Chien-Shiung Institute of Technology. The Natural Science Foundation of Jiangsu Province (BK20220301). Jiangsu Province high level innovation and entrepreneurship talent introduction plan (JSSCBS20221009). Natural Science Foundation of the Jiangsu Higher Education Institutions (25KJB320013). Science Projects of Taicang City, China (TC2022JC29). Qing Lan Project (Su Teacher [2024]14). Start-up fund for New Ph.D. Researchers of Suzhou Chien-Shiung Institute of Technology. The funders had no role in study design, data collection and analysis, decision to publish, or preparation of the manuscript.

**Competing interests:** The authors have declared that no competing interests exist.

with two additional novel subtypes recently identified [4,5]. Different subtypes exhibit distinct geographical distribution patterns and host specificities [6]. Subtypes A and B are the most widely distributed aMPV lineages globally, although their regional prevalence varies and other subtypes may predominate in specific geographic areas.[7].

In recent years, the epidemiological characteristics of aMPV have undergone dynamic changes. For instance, in the United States, following the prior elimination of subtype C, the reintroduction and subsequent spread of subtypes A and B have been detected, representing a notable epidemiological shift in this region [8]. The emergence of Avian metapneumovirus subgroup A (aMPV/A) and Avian metapneumovirus subgroup B (aMPV/B) in the U.S. led to severe outbreaks of respiratory diseases and significant economic losses in the poultry industry from late 2023 to early 2024 [9]. aMPV/A is also circulating in commercial chicken flocks in Mexico, and its complete genomic sequence has been identified using non-targeted next-generation sequencing technology [10]. This finding suggests that aMPV/A may have a broader prevalence in North America.

Vaccination remains one of the most effective strategies for controlling aMPV infections [11]. Currently available vaccines mainly include live attenuated vaccines and inactivated vaccines. For example, the aMPV/B vaccine strains BR/1890/E1/19 and BR/1891/E2/19 from Brazil have been utilized for vaccine production [12]. Novel live attenuated aMPV/B vaccines have demonstrated favorable protective efficacy in chicken flocks [11]. However, the genetic and antigenic diversity of aMPV (encompassing subtypes A, B, C, and D) and the limited cross-protection among different subtypes present significant challenges to vaccine development and selection [6]. As a paramyxovirus, the ability to manipulate the aMPV genome is crucial for investigating viral replication, pathogenic mechanisms, and developing novel vaccines [13].

The reverse genetic system is a powerful tool that enables researchers to manipulate cloned viral genomic cDNA in vitro and subsequently recover infectious viruses from the modified DNA. This technology facilitates the study of specific mutations, gene deletions, or insertions in the viral genome, thereby providing insights into viral biological characteristics [14]. For example, via the reverse genetic system, the impact of mutations in the fusion protein cleavage site of avian paramyxovirus (APMV) on viral virulence in chickens has been evaluated [15]. Additionally, although establishing a reverse genetic system for segmented RNA viruses such as rotavirus is technically challenging, optimization of this system has enabled the successful generation of recombinant rotaviruses expressing fluorescent proteins, offering new tools for viral research [16]. In aMPV reverse genetic studies, researchers have successfully established a reverse genetic system for the aMPV/B subtype LN16-A strain based on T7 RNA polymerase [17]. This system was constructed by inserting the full-length cDNA of the LN16-A strain between the T7 promoter and hepatitis D virus ribozyme, followed by co-transfection with helper plasmids encoding the viral N, P, M2-1, and L proteins, resulting in the recovery of infectious recombinant viruses [17]. Previously, other studies have also successfully rescued infectious aMPV/C viruses from cDNA and utilized this system to investigate the effects of SH gene and G protein mutations on the protective efficacy of live vaccine candidate strains [18].

Optimizing the insertion site of exogenous genes is critical for utilizing viruses as vaccine vectors or gene therapy tools. Studies have explored the capacity of aMPV, a potential viral vector, to accept exogenous gene insertions. For example, Falchieri et al identified GFP insertion sites that support gene expression while maintaining viral viability in vitro by inserting the GFP gene at all gene junctions of aMPV and constructing a full-length aMPV DNA clone in a cassette [19]. This approach aimed to evaluate the feasibility of aMPV as a vector for delivering the IBV QX gene [19]. APMV is also commonly used as a vaccine vector, capable of expressing Ebola virus glycoprotein and inducing mucosal and humoral immune responses in guinea pigs [20].

The insertion of exogenous genes into the viral vector genome requires careful consideration of its impacts on viral replication, stability, and gene expression. Current research challenges include further improving the efficiency and stability of reverse genetic systems, as well as optimizing exogenous gene insertion strategies to achieve high-level expression without compromising the biological characteristics of the viral vector [21]. For example, for Middle East Respiratory Syndrome Coronavirus (MERS-CoV), researchers have developed a novel reverse genetic system using Red-mediated recombinant cloning, which involves inserting a full-length viral genomic cDNA clone into a bacterial artificial chromosome (BAC) for maintenance to study transgene expression in the MERS-CoV genome [22]. Additionally, some studies have indicated that the protective capacity of aMPV can be enhanced by modifying its F, G, and SH genes, and found that the deletion of charged amino acids in the F protein has a more significant impact on inducing protective effects than the deletion of SH or G genes [23].

Prior studies have identified optimal exogenous gene insertion sites and established reverse genetics systems for aMPV/C subtypes, whereas no systematic investigation has been conducted on insertion site selection for aMPV/A—a subtype that has recently re-emerged in North America (USA and Mexico) and caused large-scale outbreaks in the poultry industry [8–10]. To address this gap, the present study constructed a pBR322-T7 RNA polymerase dual-based aMPV/A reverse genetics system and systematically evaluated seven intergenic insertion sites, representing the first report on the optimal exogenous gene insertion locus for aMPV/A to date.

## 2. Materials and methods

### 2.1. Cells

BSR T7/5 (baby hamster kidney cells constitutively expressing T7 RNA polymerase) and Vero cells were maintained in our laboratory. All cells were cultured in Dulbecco's Modified Eagle Medium (DMEM, Gibco, Grand Island, USA) containing 10% fetal bovine serum (FBS) and a cocktail of antibiotics (100 U/ml penicillin, 100 µg/ml streptomycin, 0.25 mg/ml amphotericin B, Thermo Scientific) at 37 °C with 5% $CO_2$ in a humidified incubator. All cell cultures and virus experiments are conducted in a controlled laboratory environment.

### 2.2. Construction of recombinant virus and auxiliary plasmids

**2.2.1. Virus and nucleic acid information.** The aMPV/A strain (GenBank accession number: PV067038) full-length genomic sequence was chemically synthesized by Sangon Biotech Company (Shanghai, China). The pcDNA3.1-EGFP plasmid (EGFP gene length: 720 bp) and empty pBR322 vector (3948 bp) were preserved in our laboratory.

**2.2.2. Genome design and Rule of Six compliance.** All recombinant aMPV/A genomes were designed to strictly follow the paramyxovirus Rule of Six; the EGFP gene (720 bp, a multiple of six) was inserted into intergenic regions with no additional nucleotide deletion/insertion, and the final full-length genomic lengths of recombinant viruses were all multiples of six (parental aMPV/A genome: 13,302 bp; recombinant aMPV/A-EGFP genomes: 14,022 bp).

**2.2.3. Gene start/stop and intergenic sequence details.** The EGFP gene was flanked by the aMPV/A canonical gene start (GS: 5'- ggacaagtcaaa-3') and gene stop (GS: 5'-ttatgaaaaaaa-3') signals, and inserted into the P-M gene spacer region of aMPV/A.

**2.2.4. Cloning strategy details.** The aMPV/A full-length genome was fragmented into 4 overlapping fragments (F1: 1–3500 bp, F2: 3450–7000 bp, F3: 6950–10500 bp, F4: 10450–13302 bp) and cloned into the pBR322 vector via BamHI/ EcoRI restriction sites; the EGFP gene was inserted into each intergenic region via In-Fusion Cloning (Takara, Japan) with specific primers (S1 Table).

**2.2.5. Plasmid construction supplement.** Helper plasmids pcN (aMPV/A N gene), pcP (aMPV/A P gene) and pcL (aMPV/A L gene) were constructed by cloning the corresponding genes into the pcDNA3.1 vector via XhoI/XbaI restriction sites; all recombinant plasmids were verified by Sanger sequencing with 100% nucleotide sequence integrity Fig 1.

## 2.3. Transfection

BSR T7/5 cells were transfected with 2 µg of raMPV/A or raMPV/A-EGFP, 0.2 µg each of the pcN and pcP plasmids, and 0.1 µg of the pcL plasmid. The transfected cells were then further incubated for 72 h in a thermostatic incubator at 37 °C with 5% $CO_2$. Cells not transfected with the helper plasmids (pcN, pcP and pcL) were set as the negative control group. During the entire transfection process, EGFP expression was dynamically monitored and photographed using an inverted fluorescence microscope (Leica, Wetzlar, Germany). At 72 h post-transfection, when approximately 80% of the cells exhibited cytopathic effects (CPE), the cell cultures were subjected to three consecutive freeze-thaw cycles and then stored at −80 °C.

## 2.4. Replication capacity of recombinant virus

Recombinant viruses were subjected to 10-fold serial dilution, and 100 µL of each dilution was inoculated onto Vero cells for viral titer quantification [24]. Viral titers were calculated using the Reed-Muench method and expressed as 50% tissue culture infective dose ($TCID_{50}$) [25]. Vero cells were subsequently inoculated with the recombinant viruses at a multiplicity of infection (MOI) of 0.1. Infected monolayer cells were harvested at 4 h intervals post-inoculation, and the mean viral titer at each time point was calculated and expressed as $\log_{10} TCID_{50}$/mL.

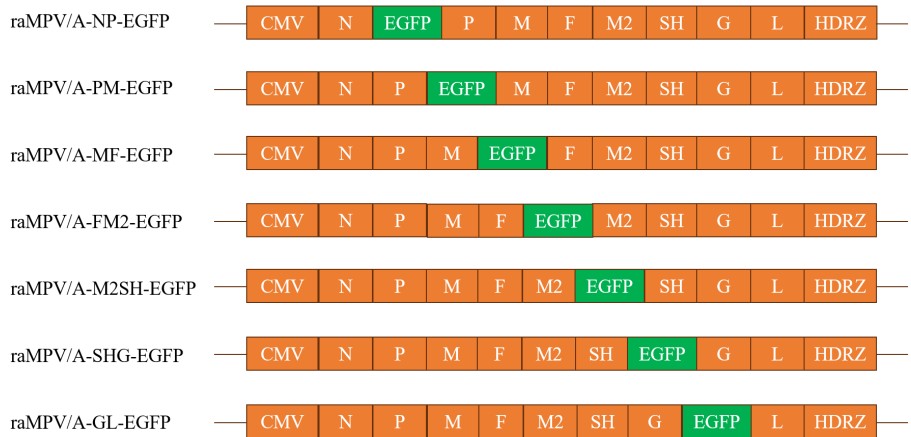

**Fig 1. Schematic diagram of aMPV/A-EGFP recombinant virus construction.** This schematic illustrates the construction of recombinant avian metapneumovirus subgroup A (aMPV/A) strains with enhanced green fluorescent protein (EGFP) inserted into seven distinct intergenic regions of the viral genome. CMV = cytomegalovirus promoter; HDRZ = hepatitis delta virus ribozyme sequence. The recombinant viruses were designated as raMPV/A-NP-EGFP (EGFP inserted into the N-P intergenic region), raMPV/A-PM-EGFP **(P-M)**, raMPV/A-MF-EGFP **(M-F)**, raMPV/A-FM2-EGFP (F-M2), raMPV/A-M2SH-EGFP (M2-SH), raMPV/A-SHG-EGFP (SH-G), and raMPV/A-GL-EGFP **(G-L)**, respectively.

### 2.5. Analysis of exogenous gene expression

Vero cells were inoculated with the seven recombinant raMPV/A-EGFP viruses at a multiplicity of infection (MOI) of 0.1. EGFP expression in each group was visualized and photographed at 48 hours post-inoculation (hpi), respectively. Fluorescence intensity was quantified using ImageJ software, and statistical analyses were conducted to compare EGFP expression levels among the recombinant strains.

### 2.6. Genetic stability of recombinant virus

Western blotting (WB) and indirect immunofluorescence assay (IFA) were employed to detect the expression of aMPV-F protein and EGFP, so as to evaluate the genetic stability of the recombinant viruses at different passages. Detailed procedures were performed as described in the aforementioned references [26].

### 2.7. Statistical analysis

All experiments were conducted with at least 3 biological replicates. Data are presented as the mean ± standard error of the mean (SEM). Statistical analyses were performed using SPSS 11.5 for Windows (SPSS Inc., Chicago, IL, USA) and GraphPad Prism 9.0 software. One-way analysis of variance (ANOVA) was applied to analyze all data for determining differences among groups. A $P < 0.05$ was considered statistically significant between groups in this study.

### 2.8. Experimental unit and replication design

In all statistical analyses throughout the study, n represents the number of biological replicates. All experiments in this study were conducted with n ≥ 3 biological replicates. The viral growth curve was obtained by repeated measurements of the same virus strain at different time points, and repeated-measures one-way ANOVA was used for statistical analysis to avoid pseudoreplication. This study was an in vitro controlled cell experiment, and all data were based on independent biological replicates of cell samples.

## 3. Results

### 3.1. The replication capacity of recombinant virus

Growth-curve analysis demonstrated that, compared with the parental strain, the titers of all recombinant viruses were slightly decreased. Specifically, raMPV/A-NP-EGFP exhibited a significantly lower viral titer than the other recombinant strains at 24–48 h ($p < 0.05$), whereas raMPV/A-GL-EGFP showed a significantly higher viral titer than the other recombinant strains at 48–72 h ($p < 0.05$) (Fig 2).

### 3.2. Analysis of exogenous gene expression

Comparison of the fluorescence intensities among these seven recombinant viruses revealed that the fluorescence intensity of raMPV/A-PM-EGFP was statistically significantly higher than that of the other recombinant strains ($P < 0.05$), whereas that of raMPV/A-GL-EGFP was statistically significantly lower than that of the other recombinant strains ($P < 0.05$) (Fig 3).

### 3.3. The genetic stability of recombinant virus

Total RNA was extracted from recombinant viruses at different passages and subjected to Sanger sequencing, which demonstrated that the EGFP gene exhibited no mutations. Consistent with this, IFA and WB results indicated that all passages of raMPV/A-EGFP were immunoreactive with the aMPV-F monoclonal antibody (Fig 4).

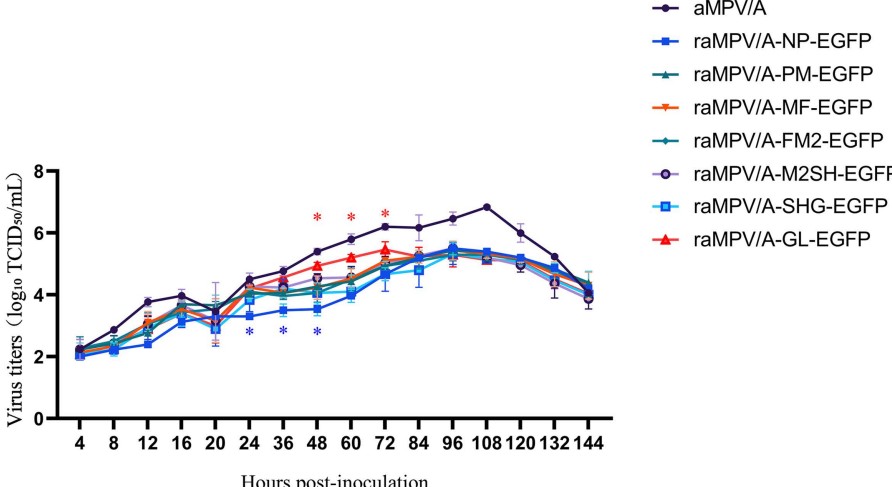

**Fig 2. The growth curve of the aMPV/A-EGFP recombinant virus.** Growth kinetics analysis of the parental aMPV/A strain and seven EGFP-expressing recombinant aMPV/A strains in Vero cells. The x-axis represents hours post-inoculation (hpi) with the virus, and the y-axis indicates viral titers expressed as $\log_{10}$ 50% tissue culture infective dose per milliliter ($\log_{10}$ TCID$_{50}$/mL). Vero cells were inoculated with each virus at a multiplicity of infection (MOI) of 0.1, and viral titers were determined at 4-hour intervals using the Reed-Muench method. Data are presented as the mean values of at least three independent experiments. $P < 0.05$ indicates a statistically significant difference compared with other recombinant strains.

## 4. Discussion

This study successfully established a reverse genetics system for aMPV/A and systematically evaluated the effects of exogenous gene insertion at different genomic loci on the replication capacity and gene expression levels of recombinant viruses. These findings provide critical data and a theoretical basis for developing aMPV as a vaccine vector or gene delivery tool.

The reverse genetics system enables researchers to rescue infectious viruses from cDNA clones and perform site-specific modifications of the viral genome via genetic engineering techniques—an achievement unattainable with traditional virological methods. Similar to previously reported reverse genetics systems for aMPV subtypes B and C, the type A aMPV system established in this study relies on T7 RNA polymerase-driven transcription of the full-length viral genome and the assistance of accessory proteins [27,28]. This confirms the universality and efficiency of this technology across different aMPV subtypes [17].

The study conducted a comprehensive assessment of exogenous reporter gene EGFP insertion at seven distinct sites within the aMPV genome. Results demonstrated that the insertion locus exerted a significant impact on both the replication capacity of recombinant viruses and the expression level of the exogenous gene. Specifically, the recombinant virus with EGFP inserted between the N and P genes exhibited the lowest replication capacity, which may be attributed to the substantial interference of this locus with viral genome integrity and transcription/replication machinery [17,29]. The gene sequences and intergenic regions of paramyxovirus genomes typically impose strict constraints on viral gene expression and replication efficiency, with any disruption leading to reduced viral adaptability [29].

In contrast, the recombinant virus harboring EGFP between the G and L genes displayed the highest replication capacity, albeit with the lowest exogenous gene expression level. This indicates that while this insertion site has a relatively minor impact on the overall viral replication process, its genomic position (proximal to the 3' end promoter region) or the influence of transcriptional regulatory elements may result in low transcription or translation efficiency of the exogenous gene [13,14]. Paramyxovirus gene transcription generally follows a "gradient expression" pattern, wherein genes closer to the 3' end promoter region exhibit higher expression levels, whereas those near the 5' end show lower expression [30,31].

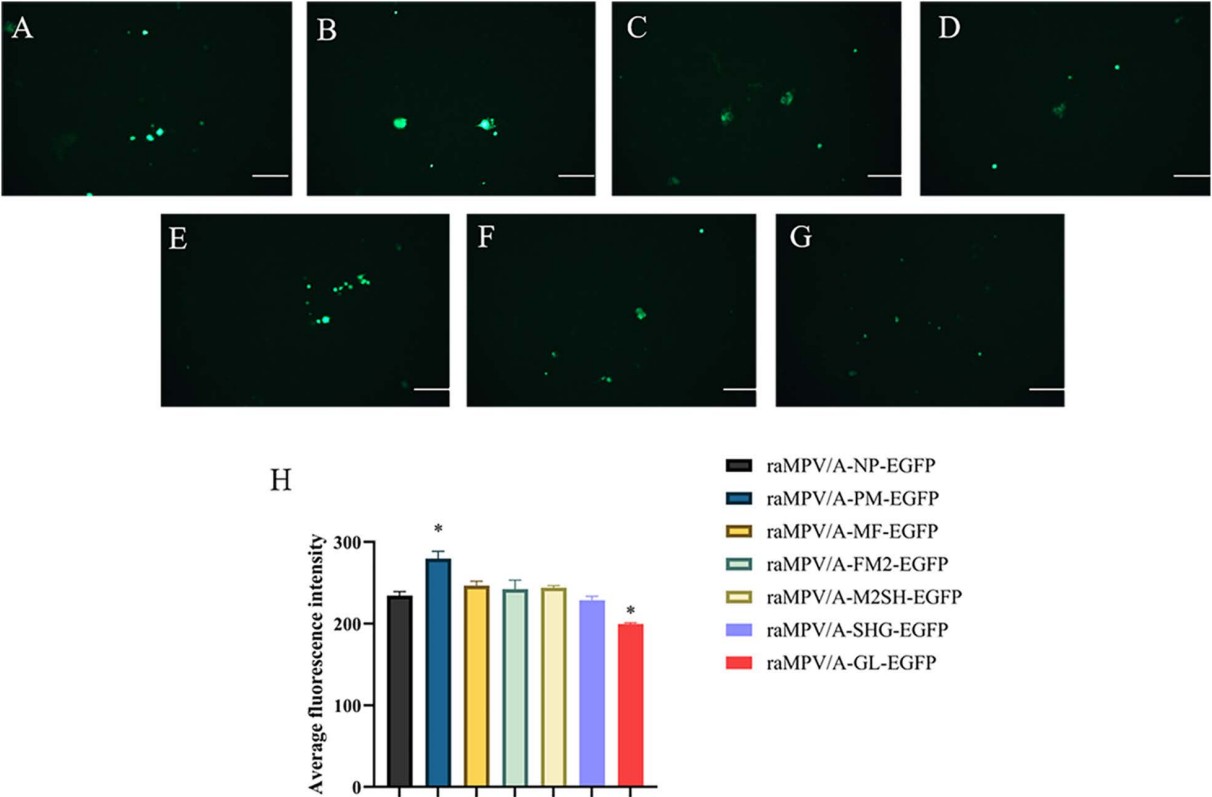

**Fig 3. Comparison of EGFP expression levels in aMPV/A-EGFP recombinant virus.** Detection of EGFP expression in Vero cells inoculated with seven aMPV/A-EGFP recombinant viruses at 48 hpi (MOI = 0.1). A–G represent fluorescence micrographs of Vero cells infected with raMPV/A-NP-EGFP **(A)**, raMPV/A-PM-EGFP **(B)**, raMPV/A-MF-EGFP **(C)**, raMPV/A-FM2-EGFP **(D)**, raMPV/A-M2SH-EGFP **(E)**, raMPV/A-SHG-EGFP **(F)**, and raMPV/A-GL-EGFP **(G)**, respectively. Scale bar = 100 µm. H is the quantitative analysis of relative EGFP fluorescence intensity in infected cells; the x-axis shows different recombinant virus strains, and the y-axis represents relative fluorescence intensity (no unit). Fluorescence intensity was quantified using ImageJ software, and data are presented as the mean values of at least three independent experiments. *P<0.05* indicates a statistically significant difference in fluorescence intensity compared with other recombinant strains.

Given that the G-L locus is relatively proximal to the 5' end, the expression level of the exogenous gene may be correspondingly limited.

Most notably, the recombinant virus with EGFP inserted between the P and M genes achieved the highest exogenous gene expression level. Considering both viral replication capacity and exogenous gene expression, the P-M intergenic region was identified as the optimal insertion site for exogenous genes in type A aMPV. This finding suggests that the P-M gene region may be a relatively "tolerant" locus, permitting exogenous gene insertion without significantly impairing viral replication while effectively driving exogenous gene expression [32]. This is of great significance for developing aMPV-based recombinant vaccines or gene delivery vectors, as efficient expression of exogenous antigens is critical for inducing robust immune responses [33].

Despite identifying the P-M site as the optimal insertion locus, further in-depth research is required to elucidate the underlying molecular mechanisms. For instance, future studies could explore the detailed dynamics of viral replication and transcription following exogenous gene insertion at this site, as well as the host cell immune response to the recombinant virus [34]. Additionally, this study only utilized EGFP as a reporter gene. In subsequent work, inserting more physiologically relevant exogenous genes—such as protective antigens from other avian pathogens—would verify the universality of this locus and further evaluate the protective efficacy of recombinant aMPV as a vaccine vector [32]. Furthermore,

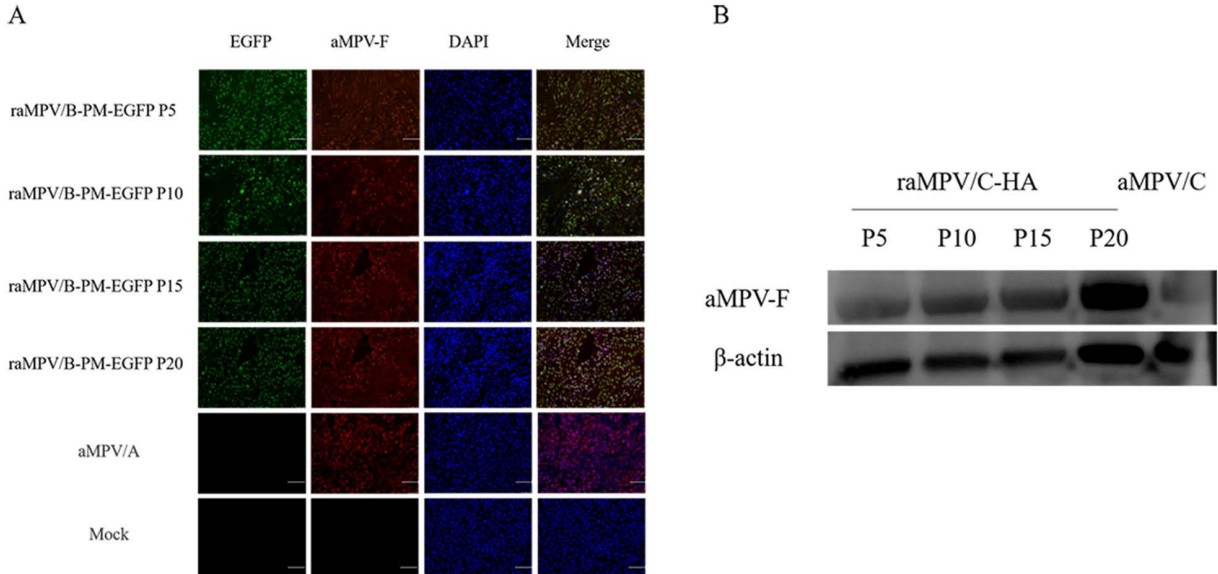

**Fig 4. Expression of EGFP in different generations of aMPV/A-PM-EGFP recombinant viruses.** Genetic stability analysis of raMPV/A-PM-EGFP (the optimal recombinant strain) after serial passage (P5, P10, P15, P20; passage 5, 10, 15, 20) in Vero cells, with parental aMPV/C as the negative control and chicken β-actin as the internal reference protein. A Indirect immunofluorescence assay (IFA) for the detection of aMPV-F protein expression in raMPV/A-PM-EGFP at different passages. Scale bar = 100 μm. B Western blotting (WB) analysis of aMPV-F protein and EGFP expression in serially passaged raMPV/A-PM-EGFP. Primary antibodies included a monoclonal antibody against aMPV-F, a specific antibody against EGFP, and a monoclonal antibody against chicken β-actin. Lanes are as follows: 1 = parental aMPV/C (negative control); 2 = raMPV/A-PM-EGFP P5; 3 = raMPV/A-PM-EGFP P10; 4 = raMPV/A-PM-EGFP P15; 5 = raMPV/A-PM-EGFP P20.

given the inherent size limitations of exogenous fragments that viral vectors can accommodate, future research should investigate the carrying capacity of the P-M site for exogenous genes of varying lengths to optimize its potential as a gene delivery platform [35].

Finally, by integrating the established reverse genetics system, future research can utilize genome editing techniques to achieve the co-expression of multiple exogenous genes – thereby enhancing the versatility of the aMPV vector.

## 5. Conclusion

In this study, a reverse genetics system for aMPV/A was successfully established and optimized. By systematically evaluating the efficiency of foreign gene insertion at seven intergenic regions, the P-M intergenic region was identified as the optimal insertion site. Recombinant viruses carrying foreign genes at this locus not only maintained robust replicative capacity and genetic stability but also achieved high-efficiency expression of the inserted foreign genes. This finding provides critical technical support for the development of aMPV/A-based vector vaccines and in-depth investigations into the molecular mechanisms of aMPV/A.

## Supporting information

**S1 Table. Primers used in this study.** The primer sequences used in the study.
(XLSX)

**S1 File. Original WB image.** The original WB images used in the research.
(PDF)

## Author contributions

**Conceptualization:** Yu Guo.

**Data curation:** Yu Guo, Ting Xue.

**Formal analysis:** Xueqiao Xie.

**Funding acquisition:** Yu Guo.

**Investigation:** Ting Xue, Xueqiao Xie.

**Methodology:** Ting Xue, Bingqing Lu, Xueqiao Xie.

**Project administration:** Ting Xue, Bingqing Lu, Huiting Wang, Yijun Zhang.

**Resources:** Yu Guo, Ting Xue, Bingqing Lu, Huiting Wang, Yijun Zhang, Jianhua Wang.

**Software:** Yu Guo, Bingqing Lu, Jiqing Zheng, Jianhua Wang.

**Supervision:** Yu Guo, Ting Xue, Bingqing Lu, Jiqing Zheng, Yirong Wang, Huiting Wang, Jianhua Wang.

**Validation:** Yu Guo, Jiqing Zheng, Yirong Wang, Huiting Wang, Yijun Zhang.

**Visualization:** Yu Guo, Jiqing Zheng, Yirong Wang, Furui Cao.

**Writing – original draft:** Yu Guo, Jiqing Zheng, Yirong Wang, Furui Cao, Jianhua Wang.

**Writing – review & editing:** Yu Guo, Furui Cao, Jianhua Wang.

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
