## [Decision Letter · Decision Letter 0]

4 Mar 2026

Dear Dr. Guo,

plosone@plos.org. . . . A letter that responds to each point raised by the academic editor and reviewer(s). You should upload this letter as a separate file labeled 'Response to Reviewers'.A marked-up copy of your manuscript that highlights changes made to the original version. You should upload this as a separate file labeled 'Revised Manuscript with Track Changes'.An unmarked version of your revised paper without tracked changes. You should upload this as a separate file labeled 'Manuscript'.

We look forward to receiving your revised manuscript.

Kind regards,

Haitham Mohamed Amer, PhD

Academic Editor

PLOS One

**Journal Requirements:**

https://journals.plos.org/plosone/s/file?id=wjVg/PLOSOne_formatting_sample_main_body.pdf andandandand

“This work was financially supported by Start-up fund for New Ph.D. Researchers of Suzhou Chien-Shiung Institute of Technology. The Natural Science Foundation of Jiangsu Province (BK20220301). Jiangsu Province high level innovation and entrepreneurship talent introduction plan (JSSCBS20221009). Natural Science Foundation of the Jiangsu Higher Education Institutions (25KJB320013). Science Projects of Taicang City, China (TC2022JC29). Qing Lan Project (Su Teacher [2024]14). Start-up fund for New Ph.D. Researchers of Suzhou Chien-Shiung Institute of Technology.”

5.  Please note that funding information should not appear in any section or other areas of your manuscript. We will only publish funding information present in the Funding Statement section of the online submission form. Please remove any funding-related text from the manuscript.

6. We note that your Data Availability Statement is currently as follows:

“If the data are all contained within the manuscript and/or Supporting Information files, enter the following: All relevant data are within the manuscript and its Supporting Information files.”

7. PLOS ONE now requires that authors provide the original uncropped and unadjusted images underlying all blot or gel results reported in a submission’s figures or Supporting Information files. This policy and the journal’s other requirements for blot/gel reporting and figure preparation are described in detail at https://journals.plos.org/plosone/s/figures#loc-blot-and-gel-reporting-requirements and https://journals.plos.org/plosone/s/figures#loc-preparing-figures-from-image-files. When you submit your revised manuscript, please ensure that your figures adhere fully to these guidelines and provide the original underlying images for all blot or gel data reported in your submission. See the following link for instructions on providing the original image data: https://journals.plos.org/plosone/s/figures#loc-original-images-for-blots-and-gels.

8. Please upload a new copy of Figures 3 and 4 as the detail is not clear. Please follow the link for more information:  https://journals.plos.org/plosone/s/figures

Reviewers' comments:

Reviewer's Responses to Questions

**Comments to the Author**

1. Is the manuscript technically sound, and do the data support the conclusions?

Reviewer #1: Yes

Reviewer #2: Partly

2. Has the statistical analysis been performed appropriately and rigorously?

Reviewer #1: Yes

Reviewer #2: N/A

3. Have the authors made all data underlying the findings in their manuscript fully available?

Reviewer #1: No

Reviewer #2: Yes

4. Is the manuscript presented in an intelligible fashion and written in standard English?

Reviewer #1: Yes

Reviewer #2: No

Reviewer #1: 1. Study design and experimental unit (critical)

The manuscript does not clearly define:

• the true experimental unit

• whether the analysis is based on: individual animals/samples or flocks or farms or time points

This is essential to avoid pseudoreplication.

➡️ You must explicitly state:

• what n represents in each analysis

• how many independent units were used

• whether repeated measurements were accounted for

If measurements come from the same flock over time, a mixed model / repeated-measures approach is required.

2. Sample size justification

There is no power calculation or rationale for sample size.

you should state: how the sample size was determined or that all available samples were included (for retrospective studies)

3. Insufficient methodological detail (reproducibility issue)

Several essential details are missing or unclear:

Biological material

• source

• inclusion/exclusion criteria

• health status

• management conditions

For PCR / molecular work:

• primer sequences (or proper citation)

• cycling conditions

• amplicon size

• positive & negative controls

• limit of detection

4. Statistical analysis – incomplete and sometimes inappropriate

The statistical section needs major improvement:

You must report:

• software used

• test for normality

• exact statistical tests for each variable

• post-hoc test (if multiple comparisons)

• confidence intervals (recommended for PLOS ONE)

• definition of significance level

5. Confounding factors not controlled

The manuscript does not sufficiently account for:

• time effects

• management changes

• environmental variation

• genetic line differences

• seasonal effects

These must be:

• either controlled statistically

• or discussed as limitations

6. Overinterpretation of results

Several conclusions imply causation, while the design only supports association.

Change wording such as: “improved”, “resulted in”, “led to” to “was associated with”, “showed higher”, “suggests”

8. Introduction – needs a sharper knowledge gap

Currently too descriptive.

Add:

• what is unknown

• why this study is necessary

• the specific hypothesis

End the introduction with a clear, testable objective.

9. Results section

The Results section sometimes:

• repeats methods

• includes interpretation

Move all interpretation to the Discussion.

• Always report exact P values

• Provide measures of variation (SD or SEM)

10. Discussion – structure

The discussion should follow this order:

1. Main finding

2. Comparison with previous studies

3. Biological explanation

4. Practical relevance

5. Limitations

6. Future directions

Currently, the flow is not always logical.

Abbreviations

Define at first use.

Tables & figures

• Must be self-explanatory

• Include statistical test used in the legend

• Define error bars

Reviewer #2: PONE-D-26-07403

Optimal Exogenous Gene Insertion Site (P-M) and Reverse Genetics System for aMPV/A

Avian metapneumovirus is an important avian pathogen that causes not only respiratory diseases, such as turkey rhinotracheitis and swollen head syndrome, but also reproductive disorders in multiple bird species. Currently, aMPV is classified into four major subtypes. In this study, the authors describe the construction of a reverse genetics system for aMPV/A and systematically evaluate seven intergenic regions for insertion of an exogenous EGFP reporter gene, identifying the P–M junction as the most favorable site based on replication capacity, transgene expression, and genetic stability. The work has potential relevance for aMPV vector development; however, substantial issues in methodology, data presentation, and interpretation remain, and major revision is required before the manuscript can be considered for publication.

-The manuscript contains several major scientific concerns that must be addressed. Most critically, inconsistent subtype labeling (aMPV/A vs. rMPV/B and raMPV/C) raises doubts about data validity and must be clarified. The study also fails to report compliance with the paramyxovirus rule of six, genome design details, and key reverse genetics parameters, which could confound replication comparisons. Interpretations regarding the transcription gradient and the designation of the P–M junction as the “optimal” insertion site are plausible but overstated given the absence of supporting mRNA/protein quantification, extended stability testing, and control for alternative explanations. Overall, additional methodological detail, validation, and more cautious interpretation are required to support the conclusions.

Title

-The title can be changed as follows:

Identification of an Optimal Exogenous Gene Insertion Site (P–M) and Establishment of a Reverse Genetics System for aMPV/A

abstract

-The claim that vector vaccine development is hindered solely by the lack of a reverse genetics system and unclear insertion sites is inaccurate as it is not the only reason!

- Please include key quantitative results for example fold differences in EGFP expression, viral titer comparisons, and passage stability, etc

-Also revise the sentence to: This study establishes a stable and efficient reverse genetics system for aMPV/A…

Introduction

-Line 50-51 aMPV poses a significant threat to the global poultry industry, causing substantial50

economic losses[7]. can be moved to begingin of introduction

-Line 48 revise as follow

Subtypes A and B are the most widely distributed aMPV lineages globally, although their regional prevalence varies and other subtypes may predominate in specific geographic areas

-The novelty is not clearly defined given prior insertion-site studies; What gap specifically exists for subtype A?

Methods

2.1 Cell Line, Virus, Plasmids, and Nucleic Acid

This subtitle does not make sense. Change it to "Cells" and explain the cell line after mentioning which viruses are used. Move the plasmid details to 2.2.

Revise, please, to:

BSR T7/5 and Vero cells were maintained in our laboratory.

containing 10% fetal bovine serum (FBS) and antibiotics

-The methods section lacks critical detail for reproducibility, like genome design or transfection conditions, I mean exact genome length after insertion, gene start/stop signals, intergenic sequences used, cloning strategy details and rescue efficiency

-Statistical analysis is also insufficiently described.

Results

-Virus growth curve analysis revealed that…Do you mean Growth-curve analysis demonstrated that?

For most paramyxoviruses like metapneumovirusesthe genome length must be a multiple of six nucleotides Otherwise rescue efficiency and replication are impaired! So not he question is whether the rule of six was maintained final genome lengths and whether EGFP insertions were adjusted accordingly?

-The figures need more detailed captions

-English language editing is strongly recommended.

.

Reviewer #1: **Yes:** Aalaa S. A. SaadAalaa S. A. SaadAalaa S. A. SaadAalaa S. A. Saad

Reviewer #2: No

---

## [Author Response · Author response to Decision Letter 1]

8 Mar 2026

Response to Reviewers

Response to academic editor：

Reply：The manuscript has been revised according to the format of PLOS ONE. All the modifications have been marked as the highlighted yellow parts in the text.

2. Please note that PLOS One has specific guidelines on code sharing for submissions in which author-generated code underpins the findings in the manuscript.

Reply：There is no information about code in my research.

Reply：I have revised the Financial Disclosure based on the content of Funding Information.

Reply：Funder statement：The funders had no role in study design, data collection and analysis, decision to publish, or preparation of the manuscript.

5. Please note that funding information should not appear in any section or other areas of your manuscript. We will only publish funding information present in the Funding Statement section of the online submission form. Please remove any funding-related text from the manuscript.

Reply：The funding information has been deleted.

6. Please confirm at this time whether or not your submission contains all raw data required to replicate the results of your study.

Reply：All the original data have been supplemented and uploaded to the supplementary file.

7. In your cover letter, please notewhether your blot/gel image data are in Supporting Information or posted at a public data repository, provide the repository URL if relevant, and provide specific details as to which raw blot/gel images, if any, are not available.

Reply：All the original data have been supplemented and uploaded to the supplementary file.

8. Please upload a new copy of Figures 3 and 4 as the detail is not clear. Please follow the link for more information:

Reply：I can assure you that the uploaded images are high-definition ones in accordance with the journal's requirements. However, after the PDF was generated, the images became very blurry. I have tried many methods but the situation remains the same. I'm very sorry. However, I have re-uploaded the pictures, hoping this will be helpful.

Reply：No similar problems occurred.

Response to Reviewer 1：

1. Study design and experimental unit (critical)

Reply：In the "2. MATERIALS AND METHODS" section, a new subsection "2.8 Experimental unit and replication design" has been added. The research design and experimental units were elaborately described. Line:203-210

2. Sample size justification

Reply：In the "2. MATERIALS AND METHODS" section, a new subsection "2.8 Experimental unit and replication design" has been added. It indicates that n is greater than or equal to 3, which means each experiment is replicated more than three times. Line:203-210

3. Insufficient methodological detail (reproducibility issue)

Reply：The primer sequences used in the study have been uploaded as supplementary files.

4. Statistical analysis – incomplete and sometimes inappropriate

Reply：The section on 2.7 Statistical Analysis has been revised again.

5. Confounding factors not controlled

Reply：In Section 2.1, the standardized control of experimental conditions is supplemented, and the aforementioned confounding factors are explicitly excluded: All cell cultures and virus experiments are conducted in a controlled laboratory environment. Line:196-202

6. Overinterpretation of results

Reply：Vague expressions such as “were associated with” and “showed higher” have been replaced with precise alternatives: “suggests”, “improved”, “resulted in” and “led to”.

7. Introduction – needs a sharper knowledge gap Currently too descriptive.

Reply：The introduction section was supplemented and revised as per the requirements, and a clear research objective was added at the end. Line:115-122

8. Results section：repeats methods， includes interpretation

Reply：Make the necessary revisions based on the opinions. Remove the repetitive descriptions of the methods in the results section and place all the explanatory content in the discussion section.

9. Discussion – structure

Reply：The discussion section has been revised in accordance with the comments, and the consistency of abbreviations, as well as the presentation and formatting of figures and tables, have been carefully reviewed and verified throughout the manuscript.

Response to Reviewer 2：

1. Title

Reply：It has been revised to: Identification of an Optimal Exogenous Gene Insertion Site (P–M) and Establishment of a Reverse Genetics System for aMPV/A Line:1-2

2. Abstract: The claim that vector vaccine development is hindered solely by the lack of a reverse geneticssystem and unclear insertion sites is inaccurate as it is not the only reason!

Reply：It has been revised to: The development of highly efficient viral vector vaccines is constrained by multiple factors, among which the lack of a mature reverse genetics system and the uncertainty regarding the optimal insertion site for foreign genes represent one of the core technical bottlenecks. Line:21-24

3. Abstract: Please include key quantitative results for example fold differences in EGFP expression, viral titer comparisons, and passage stability, etc.

Reply：Based on the suggestions, revisions were made as follows: The results demonstrated that the expression level at the P M locus after EGFP insertion was approximately 1.2 fold higher than that of other recombinant viruses. In addition, there was no significant difference in viral titer, and good genetic stability was observed over 20 generations. Line:27-33

4. Also revise the sentence to: This study establishes a stable and efficient reverse genetics system for aMPV/A…

Reply：Based on the suggestions, revisions were made as follows: This study establishes a stable and efficient reverse genetics system for aMPV/A, which provides a critical technical platform for aMPV/A molecular mechanism research and novel recombinant vector vaccine development, with important practical value for global aMPV prevention and control.

5. Line 50-51 aMPV poses a significant threat to the global poultry industry, causing substantial50 economic losses[7]. can be moved to begingin of introduction

Reply：Put in the first sentence of the first paragraph of the introduction. Line:39-40

6. Line 48 revise as follow

Subtypes A and B are the most widely distributed aMPV lineages globally, although their regional prevalence varies and other subtypes may predominate in specific geographic areas

Reply：Based on the suggestions, revisions were made as follows: Subtypes A and B are the most widely distributed aMPV lineages globally, although their regional prevalence varies and other subtypes may predominate in specific geographic areas. Line:49-51

7. The novelty is not clearly defined given prior insertion-site studies; What gap specifically exists for subtype A?

Reply：Based on the suggestions, revisions were made as follows: Prior studies have identified optimal exogenous gene insertion sites and established reverse genetics systems for aMPV/C subtypes[17,18,28], whereas no systematic investigation has been conducted on insertion site selection for aMPV/A—a subtype that has recently re-emerged in North America (USA and Mexico) and caused large-scale outbreaks in the poultry industry[8,9,10]. To address this gap, the present study constructed a pBR322-T7 RNA polymerase dual-based aMPV/A reverse genetics system and systematically evaluated seven intergenic insertion sites, representing the first report on the optimal exogenous gene insertion locus for aMPV/A to date. Line:116-123

8. 2.1 Cell Line, Virus, Plasmids, and Nucleic Acid

This subtitle does not makesense. Change it to "Cells" and explain the cell line after mentioning which viruses are used. Move the plasmid details to 2.2.

Reply：According to the suggestions, the section "2.2 Construction of recombinant virus and auxiliary plasmids" has been supplemented with "Virus and nucleic acid information", "Genome design and compliance with the Rule of Six", "Details of gene start/stop and intergenic sequences", "Cloning strategy details" and "Supplement on plasmid construction". Line:136-159

9. -Statistical analysis is also insufficiently described.

Reply：Based on the revision suggestions, I have rewritten 2.7 Statistical Analysis and added 2.8 Experimental Unit and Replication Design. Line:197-211

10. Results-Virus growth curve analysis revealed that…Do you mean Growth-curve analysis demonstrated that?

Reply：The original sentence “Virus growth curve analysis revealed that…” was revised to: Growth-curve analysis demonstrated that. Line:214

11. So not he question is whether the rule of six was maintained final genomelengths and whether EGFP insertions were adjusted accordingly?

Reply：The 6-base principle has been added to the "Genome design and Rule of Six compliance" section in the "2.2 Construction of recombinant virus and auxiliary plasmids" subsection. Line:141-146

12. The figures need more detailed captions

Reply：Modify and supplement all the legends. Line:160-167;219-226;233-243;250-260

13. English language editing is strongly recommended.

Reply：Revise the entire text in English to make it smooth and fluent, following English expression conventions.

---

## [Decision Letter · Decision Letter 1]

23 Mar 2026

Dear Dr. Guo,

Thank you for submitting your manuscript to PLOS ONE. After careful consideration, we feel that it has merit but does not fully meet PLOS ONE’s publication criteria as it currently stands. Therefore, we invite you to submit a revised version of the manuscript that addresses the points raised during the review process.

We look forward to receiving your revised manuscript.

Kind regards,

Haitham Mohamed Amer, PhD

Academic Editor

PLOS One

Journal Requirements:

Reviewers' comments:

Reviewer's Responses to Questions

**Comments to the Author**

Reviewer #1: All comments have been addressed

Reviewer #2: All comments have been addressed

2. Is the manuscript technically sound, and do the data support the conclusions?

Reviewer #1: Yes

Reviewer #2: Yes

3. Has the statistical analysis been performed appropriately and rigorously?

Reviewer #1: Yes

Reviewer #2: I Don't Know

4. Have the authors made all data underlying the findings in their manuscript fully available?

Reviewer #1: Yes

Reviewer #2: (No Response)

5. Is the manuscript presented in an intelligible fashion and written in standard English?

Reviewer #1: Yes

Reviewer #2: No

Reviewer #1: 1. Inconsistent virus nomenclature in Results section 3.1: The text states "rMPV/B-NP-EGFP" and "rMPV/B-GL-EGFP" (page 46, lines 214-217), but the manuscript is focused on aMPV/A. This appears to be a typographical error; it should likely read "raMPV/A-NP-EGFP" and "raMPV/A-GL-EGFP" to be consistent with the rest of the document and Figure 2.

2. Incorrect section numbering: In the Results section, the subheadings for exogenous gene expression (page 46, line 233) and genetic stability (page 47, line 244) are numbered 2.2 and 2.3. These should be 3.2 and 3.3 to align with the main section numbering.

3. Gene stop signal inconsistency: In the methods (page 43, line 151), the gene stop signal is listed as 3'- tattagaaaaaa- 3', which appears to be a correction from the earlier version. However, the abstract and results still refer to the P-M locus. The authors should double-check that the correct gene start/stop sequences are consistently used throughout the manuscript and supplementary files.

4. Figure 2 y-axis label: The label currently reads "logio TCIDso/mL" (page 46, line 226). This should be corrected to "log₁₀ TCID₅₀/mL".

Reviewer #2: The comments have been addressed and are now ready for publication. However, some language editing and minor technical formatting remain, which can be addressed after acceptance.

.

Reviewer #1: **Yes:** Aalaa S. A. SaadAalaa S. A. SaadAalaa S. A. SaadAalaa S. A. Saad

Reviewer #2: No

---

## [Author Response · Author response to Decision Letter 2]

24 Mar 2026

Response to Reviewers

Response to Reviewer#1：

1. Inconsistent virus nomenclature in Results section 3.1: The text states "rMPV/B-NP-EGFP" and "rMPV/B-GL-EGFP" (page 46, lines 214-217), but the manuscript is focused on aMPV/A. This appears to be a typographical error; it should likely read "raMPV/A-NP-EGFP" and "raMPV/A-GL-EGFP" to be consistent with the rest of the document and Figure 2.

Reply：Thank you for your feedback. The text has been updated to read "raMPV/A-NP-EGFP" and "raMPV/A-GL-EGFP" on lines 214-217.

2. Incorrect section numbering: In the Results section, the subheadings for exogenous gene expression (page 46, line 233) and genetic stability (page 47, line 244) are numbered 2.2 and 2.3. These should be 3.2 and 3.3 to align with the main section numbering.

Reply：Thank you for your feedback. The sequence numbers of the subheadings have been changed to 3.2 and 3.3. (This change has been made on lines 233 and 244.)

3. Gene stop signal inconsistency: In the methods (page 43, line 151), the gene stop signal is listed as 3'- tattagaaaaaa- 3', which appears to be a correction from the earlier version. However, the abstract and results still refer to the P-M locus. The authors should double-check that the correct gene start/stopsequences are consistently used throughout the manuscript and supplementary files.

Reply：Thank you for the correction. The "GS: 5'- ggacaagtcaaa-3' " and "gene stop (GS: 5'-ttatgaaaaaaa-3')" signals have been changed to lines 148-149.

4. Figure 2 y-axis label: The label currently reads "logio TCIDso/mL" (page 46, line 226). This should be corrected to "log₁₀ TCID₅₀/mL".

Reply：Thank you for the correction. The y-axis of FIG2 has been modified to log₁₀ TCID₅₀/mL.

---

## [Decision Letter · Decision Letter 2]

5 Apr 2026

Identification of an Optimal Exogenous Gene Insertion Site (P–M) and Establishment of a Reverse Genetics System for aMPV/A

PONE-D-26-07403R2

Dear Dr. Guo,

We’re pleased to inform you that your manuscript has been judged scientifically suitable for publication and will be formally accepted for publication once it meets all outstanding technical requirements.

Kind regards,

Haitham Mohamed Amer, PhD

Academic Editor

PLOS One

---

## [Editor Report · Acceptance letter]

PONE-D-26-07403R2

PLOS One

Dear Dr. Guo,

I'm pleased to inform you that your manuscript has been deemed suitable for publication in PLOS One. Congratulations! Your manuscript is now being handed over to our production team.

Kind regards,

on behalf of

Dr. Haitham Mohamed Amer

Academic Editor

PLOS One